# Fragmentation Mechanism in a Nitrogen Dielectric Barrier Discharge Plasma on Fluoropolymer Polymer Films

**DOI:** 10.3390/ma16030942

**Published:** 2023-01-19

**Authors:** Faegheh Fotouhiardakani, Morgane Laurent, Jacopo Profili, Sethumadhavan Ravichandran, Gowri Dorairaju, Gaetan Laroche

**Affiliations:** 1Laboratoire d’Ingénierie de Surface, Centre de Recherche sur les Matériaux Avancés, Département de Génie des Mines, de la Métallurgie et des Matériaux, Université Laval, Québec, QC G1V 0A6, Canada; 2Centre de Recherche du CHU de Québec, Hôpital St François d’Assise, 10 rue de L’Espinay, Québec, QC G1L 3L5, Canada; 3Saint-Gobain Research North America, 9 Goddard Rd, Northborough, MA 01532, USA

**Keywords:** plasma, fluoropolymer, surface treatment, hydrophilicity, coating, dielectric barrier discharge

## Abstract

Due to their chemical inertness and low friction coefficient, fluoropolymers are today widely employed in sectors of activity as diverse and distinct as the textile industry, architectural sector, and medicine. However, their low surface energy results in poor adhesion, for example, when used for a component in a composite device with multiple other materials. Among the techniques used to enhance their adhesion, atmospheric pressure discharges provide a fast and low-cost method with a reduced environmental impact. Although this approach has proven to be efficient, the different chemical and physical processes in the discharge remain not fully understood. In this study, fluoropolymer surfaces were modified using an atmospheric pressure dielectric barrier discharge in a nitrogen and organic precursor environment. To prevent any damage to fluoropolymer surfaces, the dissipated power in the discharges was tuned by applying a duty cycle. Evidence shows that plasma treatment allows for the incorporation of oxygen and nitrogen in the surface resulting in the formation of hydrophilic functionalities such as carbonyl groups both in ketone and amide form, amine, and hydroxyl groups after 180 s of treatment. Overall, the data reveal that the discharge duty cycle has more effect on the oxygen and carbon content in the coating than the precursor concentration. In addition, increasing the precursor concentration limits the molecular fragmentation and nitrogen incorporation into the coating. These experiments enable the building of a better fundamental understanding of the formation mechanism of such chemical moieties at the fluoropolymer surface.

## 1. Introduction

Fluoropolymers, such as ethylene tetrafluoroethylene (ETFE), polytetrafluoro-ethylene (PTFE), perfluoroalkoxy polymer (PFA), and fluorinated ethylene-propylene (FEP), are mechanically strong materials due to the presence of carbon–carbon (~347 kJ mol^−1^) and carbon–fluorine bonds (~485 kJ·mol^−1^) in their backbones. In addition, fluoropolymers exhibit chemical inertness, low surface energy, and show excellent dielectric properties [1]. Thanks to these unique features, research in the field of fluoropolymers has attracted growing interest over the years, for numerous potential applications, including textiles, architectural, electronics, and biomedical devices [2]. The low surface energy of these polymers usually results in low adhesion and non-sticking properties [3]. Although their anti-adhesive properties present an excellent advantage for many applications, this limits their use in composite due to their inability to bond to other materials. Plasma treatment is a solvent-free and dry method that enables the modification of the surface of materials without altering their bulk properties [4,5,6]. In addition, plasma has been widely used to modify polymer surfaces in different sectors, such as printing, labeling, and packaging [7,8]. In comparison to low-pressure plasma systems, atmospheric pressure plasma (APP) reactors can minimize process costs by eliminating the requirement for vacuum equipment, allowing for faster treatment times and the ability to perform continuous treatments. Therefore, this process is fast and cost-effective, which makes it suitable for high throughput manufacturing environments [9]. Despite the various advantages presented here, two main challenges usually limit the use of plasma in the industry. First, the plasma modification induced on the polymer experiences aging which changes the surface chemical structure over time. Second, the use of atmospheric pressure discharge often involves highly energetic localized filaments [10,11]. These phenomena lead to an inhomogeneous modification of the surface and/or could induce damage to the sample due to localized heat transfer [12]. For this reason, additional studies are necessary to improve the use of APP approaches and their understanding. A dielectric barrier discharge (DBD) configuration consists of two metallic electrodes with at least one of the electrodes being covered by a dielectric. In this research, the fluoropolymer film acts as the dielectric layer in the DBD configuration. In this configuration, a non-polymerizable gas mixture can be used to enhance the adhesion by breaking molecular bonds on the film surface and enabling recombination with free radicals from the discharge to form polar functionalities [13]. A non-polymerized gas is a compound that does not polymerize yet produces chemical functionalities on the surfaces when combined with reactive gases such as H_2_ and N_2_. In applications involving adhesion, mixing a reactive gas with a non-polymerize precursor might result in greater crosslinking [14,15,16]. Different parameters (dilution gas, thin-film precursor, additives) can affect the discharge regime and behavior [17,18]. These parameters also influence the chemical composition of the coating and hence, its physico-chemical characteristics [19]. For example, using N_2_ as the primary gas allows the formation of nitrogen-containing functionalities. In this context, the use of an organic precursor such as acrylic acid, allyl alcohol, ethanol, and allylamine [20] make it possible to form a hydrophilic layer on the surface with various functional polar groups, such as amines (NH_2_), amides (N-C=O), and carboxyls (COO) [21,22]. One of the critical parameters in plasma is the input power to control monomer interactions with energetic species [23,24]. Savage et al. [24] used pulsed wave (PW) to reduce the fragmentation of monomers. Pulsed plasma can improve the control of thin film chemistry by controlling energy dissipation. In this work, a duty cycle (DC) was used to pulse the discharge. The DC is equal to the t_on_/(t_on_ + t_off_) where t_on_ corresponds to the time when the plasma species are produced and molecular fragmentation occurs, while t_off_ refers to time when the plasma is off [25]. The surfaces of fluoropolymers were coated for 180 s by an atmospheric pressure DBD using N_2_ and a non-polymerizable organic precursor containing oxygen in static mode to form a resistant thin film coating. The coating is characterized as a function of precursor concentration and duty cycle (to raise the power level). The thin film was then analyzed to comprehend the molecular fragmentation and nitrogen incorporation into the precursor.

## 2. Experimental Section

### 2.1. Materials

ETFE films 0.127 mm in thickness (supplied by Saint-Gobain Research North America, Northborough, MA, USA) were treated in static mode for 180 s in a DBD system. The carrier gas used was nitrogen, and an organic oxygen-containing precursor was chosen for its capacity to form polar bonds after plasma deposition.

### 2.2. Plasma Reactor

The discharge configuration used in this work is depicted in Figure 1. The experimental setup consisted of a power supply, two flat stainless high-voltage electrodes parallel to a cylindrical ground electrode, and a vacuum system. The electrodes were placed in a stainless-steel chamber. The area of each high-voltage electrode was 0.95 cm × 14.9 cm, and the ground electrode was a stainless-steel cylinder with a 150 mm diameter. The polymer films were placed on the surface of the ground electrode facing the high-voltage electrodes. A Super Air Knife airflow laminator (EXAIR) was used to send a laminar flow of the carrier nitrogen gas and the gaseous precursor through the small space between the two top electrodes. In addition, a vacuum system was used to decrease the pressure near 0.8 to 1 kPa before each treatment to remove the gas impurities that could affect the treatment. ETFE acted as a dielectric barrier, preventing polymer damage from the spark transition.

The power supply of the discharge was built from an arbitrary function generator (AFG-2021, Tektronix, Beaverton, OR, USA) connected to an audio amplifier PL380 from QSC Audio products, LLC (Costa Mesa, CA, USA) working in the 10 Hz–100 kHz range. A transformer (RAFTabtronics, Longwood, FL, USA) was used to amplify the signal, providing an applied voltage (V_a_) up to 18 kV between the high voltage and ground electrodes with a bandwidth from 5 to 15 kHz. The applied voltage was measured at the exit of the transformer using a high-voltage probe (P6015A, 1000x attenuation, Tektronix). An additional 100 nF capacitor was inserted on the ground side after the discharge. A voltage was measured on the capacitor by a passive probe (TPP0500B, Tektronix) and provided access to the charge flowing through the discharge, enabling the calculation of the dissipated power via the Lissajous figure method [26,27]. The average power density was calculated in W.cm^−2^. Finally, a high-performance oscilloscope (Tektronix, Mixed Domain Oscilloscope MDO3054, 500 MHz, 10 M record length) was used to follow and save the current/voltage curves. Additionally, a duty cycle was used, which consisted of turning on and off the plasma by multiplying the sinusoidal signal by a squared waveform with a frequency that corresponded to the time on of the signal. This enabled the control of the discharge and avoided overheating and arc formation [13,14]. The effect of the precursor concentration and the duty cycle were studied to investigate the relationship between the fragmentation of organic molecules and the chemical composition of the coated surfaces. An alternating voltage having a frequency of 5 kHz and an amplitude of 11 kV was used. The breakdown voltage was 9 kV, while the maximum voltage before arc formation was 13 kV. Considering this, 11 kV was selected as the average voltage. The total gas flow (N_2_^+^ precursor) was set to 5 SLM (Standard Liter per Minute), and the inter-electrode gas gap was set to 1 mm (Table 1). The precursor amount varied between 3% and 15% with respect to the total flow. The precursor concentration was selected due to the limitation of the flowmeter on the maximum precursor amount possible to inject into the plasma zone for a 5 SLM total flow. The duty cycle used to regulate the power during the treatment ranged between 10% and 50%. The limit was fixed at 50% DC to prevent arc formation, polymer overheating, and surface damage. Table 1 summarizes all conditions studied in this work.

### 2.3. Characterization Techniques

#### 2.3.1. X-ray Photoelectron Spectroscopy

X-ray photoelectron spectroscopy (XPS) (PHI 5600-ci spectrometer, Physical Electronics, Chanhassen, MN, USA) was used to measure the relative atomic composition on the various investigated surfaces. The measurements were carried out with an incident angle of 45° on sample 0.5 mm^2^ area at a residual pressure of 8 × 10^−9^ Torr. A survey spectrum covering the range of 0–1200 eV was first recorded using a standard aluminum K_α_ X-ray source (1488.6 eV) at 300 W, with charge compensation (neutralizer). A monochromatic magnesium anode (1253.6 eV) was utilized at 150 W without charge neutralization for high resolution spectra of carbon (280–300 eV). Multipak (9.0 V) software was used to process XPS survey scans, and the peak area ratio used to identify the elemental composition of the ETFE sample and the coating layer. The CF_2_ component of the C1s spectrum at 291 eV was used to calibrate the energy scale for the ETFE, and the hydrocarbon component of the C1s spectrum (285.0 eV) was utilized to calibrate the energy scale for the coating. Multipak was employed to curve-fit high-resolution C1s peaks. Peaks were curve fitted using Gaussian–Lorentzian peak forms over an iterated Shirley background, with the full width at half maximum (FWHM) of each line shape kept between 0.9 and 1.2 eV. Each of the samples was subjected to four measurements to estimate the homogeneity of the treatment and the reproducibility of the measurement. Furthermore, this coating investigation was performed twice for each plasma setting to ensure that the treatment process was reproducible.

#### 2.3.2. Surface Energy

Surface energy was determined through static contact angle measurements. This was performed using a WCA 2500 XE system (AST, Billerica, MA, USA) equipped with a built-in high-resolution CCD camera. Per each analysis 0.5 µL droplet of deionized water and diiodomethane (ReagentPlus^®^, 99%, contains copper as neutralizer, Sigma-Aldrich, St. Louis, MO, USA) were placed on the surface. A software computed the angles at the edge of the droplet. Ten contact angles (five drops per liquid) were measured for each sample to get an average value and its standard deviation. Surface energy was calculated from the polar and dispersive surface energy components based on Fowkes theory using contact angle values of those liquids utilizing Equation (1) [28,29]:(1)$${\left(\sigma_{L}^{d}\right)}^{1/2}{\left(\sigma_{s}^{d}\right)}^{1/2}+{\left(\sigma_{L}^{p}\right)}^{1/2}{\left(\sigma_{s}^{p}\right)}^{1/2}=\frac{\sigma_{L}}{2}\left(\cos\theta +1\right)\;$$

#### 2.3.3. Attenuated Total Reflectance Fourier Transformed Infrared Spectroscopy (ATR-FTIR)

Fourier-transform infrared spectroscopy (FTIR-ATR) was used to identify the coating chemical functionalities. The spectra were recorded with 4 cm^−1^ of resolution on an Agilent 600 Series FTIR spectrometer (Agilent, Mulgrave, Australia). A Harrick’s SplitPea™ (Pleasantville, NY, USA) was set to ensure strong contact between sample and ATR crystal. The spectra were recorded between 4000 and 400 cm^−1^ range with 128 scans at 4 different points from a 1 cm × 1 cm treated sample. All spectra were normalized to the maximum peak intensity (1040 cm^−1^, C-F bond from ETFE [30]) to consider potentially unexpected variables/errors such as sample thickness. For each sample, four points were measured in various locations on the coating to calculate the standard deviation of the average area under the peaks assigned and study the growth mode. The curve fitting of the peaks was curve fitted using Voigt peak shape of full width-at-half-maximum (FWHM) below 50 cm^−1^.

## 3. Results and Discussion

### 3.1. Surface Composition of Plasma-Treated Samples

Figure 2 shows the concentrations of carbon and fluorine on the surface of plasma-treated ETFE samples as a function of the amount of precursor injected in the discharge for each applied duty cycle (DC). All the investigated experimental conditions lead to a decrease in the surface fluorine concentration from 55% (untreated sample) to approximately 0%, concomitant with an increase in the surface carbon concentration (Figure 2). These results are consistent with the deposition of an organic thin film on the ETFE samples. Interestingly, the carbon surface concentration was shown to be dependent on the duty cycle applied more than the concentration of precursor injected in the plasma.

For a precursor concentration of less than 10%, the amount of carbon in the thin film remained almost unchanged at approximately 72% when the duty cycle in the plasma was set at 10%. Under the same applied duty cycle, an increase in the amount of precursor from 10 to 15% made it possible to produce thin films with a slightly higher carbon content, e.g., 75%. Thin films deposited under a DC of 30% revealed a surface carbon content varying from 72 to ~78% as the concentration of precursor in the discharge went from 10 to 30%. A similar trend was observed in ETFE samples subjected to a DC of 50%. Here, the carbon surface concentration found was slightly higher, from 75 to 78%, with an increase in the amount of precursor injected in the plasma.

The results presented in Figure 2 show that for a given amount of precursor injected in the plasma, increasing levels of DC entails rising concentrations of surface carbon in the thin films. This can primarily be explained by the rise of the measured power which is associated with the increase in duty cycle. For instance, the power density rose from 0.17 W.cm^−2^ to 0.51 W.cm^−2^ and then to 0.91 W.cm^−2^ by changing the DC from 10% to 30% and to 50%, as mentioned in Table 1. Such increase in power is likely to cause an increase in the concentration of the precursor fragmentation products (rich in carbon-containing moieties) thereby depositing them on the surface of ETFE.

In addition to carbon and fluorine, small amounts of oxygen and nitrogen are also found on the surface of the ETFE samples following plasma treatment. As the precursor injected in the discharge contains carbon, oxygen, and hydrogen atoms, nitrogen found in the thin films solely comes from the carrier gas, N_2_. Figure 3a shows the O/C ratio as a function of the precursor injected in the discharge for each applied DC. For a 10% DC, plasma-treated films revealed a O/C ratio variation from 0.22 to 0.26 as the concentration of the precursor went from 3% to 10%. Interestingly, when the precursor concentration in the plasma was 15%, this ratio was found to decrease to 0.21. For 30% DC, the O/C ratio remained unchanged at approximately 0.21, for precursor concentrations ranging between 3 and 10%, and then decreased to 0.17 when the concentration of the precursor was set at 15%. For 50% DC, the O/C ratio increased from 0.14 to 0.18 for a concentration of the precursor between 3 and 10% and remained unchanged between 10 and 15%. It can be argued that an increase in the amount of the precursor injected in the plasma should result in an increasing concentration of oxygen in the coatings [17]. With duty cycles ranging from 10 to 50%, the probability of precursor bonds breaking due to collision with reactive species in the plasma increases and so the resulting molecule fragments. However, as can be deduced from Figure 3a, the available energy per molecule may limit this phenomenon [18]. XPS results indicate that low levels of applied DC favor the deposition of oxygen-containing species from the plasma phase onto the ETFE substrates. In contrast, high levels of DC appear to make the incorporation of oxygen-containing species from the plasma phase onto the ETFE substrates difficult. As a result, less oxygen is analyzed on the surface [19,20]. These results support those of Laurent et al. [31]. In their study, the O/C ratio was shown to decrease with the power dissipated in the discharge until reaching saturation, when dissipated powers as high as 2.10 W.cm^−2^ were applied in a N_2_ discharge [23]. Additionally, the carbon matrix can be strongly crosslinked when using higher DC level. [32,33]. Similar results have been reported during plasma polymer deposition varying DCs. A. Airoudj et al. [34], mentioned that an increase in the duty cycle results in a rise in Maleic anhydride molecule fragmentation. Consequently, different reaction mechanisms take place, resulting in the retention of fewer anhydride functionalities in the plasma polymer thin film structure. In addition, their XPS analysis showed that the O/C concentration ratio increased from 0.21 in the continuous wave plasma thin film to 0.31 in the pulsed plasma polymerized thin films. The authors explained that a lower DC (or average input power) formed a polymer network with more anhydride groups which was accompanied by a decrease in crosslinking. During plasma off-excitation periods, radical-monomer processes led to less cross-linked, more ‘polymer-like’, structures [32]. These observations are also supported by the work of Vallade et al. (2018) [35]. The authors modified the surface of fluoropolymers using a semi-industrial corona discharge in a DBD configuration, similar to the one used in the present study, with nitrogen and an organic precursor. In this study, the partial least square regression (PLSR) model was used to predict the relationship between the plasma experimental parameters and the physicochemical characteristics of the resulting coating. Their model demonstrated that electrical parameters had more effect on the chemical deposition as compared with other parameters such as gas flow and nitrogen vibrational temperature.

Figure 3b shows the N/C ratio as a function of the precursor injected in the discharge for each DC applied. Contrary to O/C ratios, the N/C ratio appears not to be affected by the duty cycle applied, while being mainly governed by the precursor concentration. Indeed, increasing the precursor concentration from 3 to 10% makes it possible to produce thin films with N/C ratios of 0.2 and 0.1, respectively, for all investigated duty cycle values. When the concentration of the precursor was set at 15%, the N/C remained unchanged at 0.1. As the total gas flow rate of 5 L/min was kept constant in all experiments, nitrogen concentration in the discharge was found to slightly decrease by increasing that of the precursor.

Further details on the carbon bonding environment were assessed using XPS analyses (Figure 4a,b). Figure 4a shows curve fitting of the C1s spectrum of untreated polymer (ETFE). For the untreated ETFE polymer, the C1s envelope can be resolved into three bands originating from fluorine-containing functionalities, such as CF_2_ (291 eV—highlighted as carbon n#2 in Figure 4a) and C-F (288.4 eV—shown as carbon n#3 in Figure 4a) as well as from single bonded C-C/C-H containing species that are linked to a carbon bonded with fluorine (286.5 eV—shown as carbon n#1 in Figure 4a). Following plasma treatment, the C1s envelope can still be resolved into three bands despite being shifted toward 285 eV (Figure 4b). Worthy of mention here is the absence of the C-F_2_ feature at 291 eV, which substantiates the deposition of a coating with a thickness greater than 5–10 nm (due to the C1s electrons mean free path) on the ETFE sample. The bands at 286.5 eV and 287.8 eV can be attributed to the presence of hydrophilic single-bonded C-O/C-N and double-bonded C=O/N-C=O containing species in the coatings, respectively. The C—N bonds are likely to be generated from collision of the nitrogen carrier gas with the precursor or directly with the substrate while the C—O bonds may originate from the partial dissociation of the double C=O bond in the precursor [23].

Approximately 13–28% of carbon atoms are forming C—O/C—N bonds, and 12–20% correspond to C=O functionalities (Table 2). The band at 285 eV accounts for 60–72% of the carbon bonding environment and can be attributed to C-C/C-H bonds already present in the backbone of the precursor as well as to the new bonds formed on the surface of the coating.

Table 3 shows the assignments of the main absorption peaks in the infrared spectrum of plasma-coated ETFE samples. To understand how fragmentation of the precursor may take place in the discharge, the C=O/(C-C, C-H) ratio was plotted as a function of the concentration of the precursor in the discharge for each applied DC (Figure 5). Regardless of the applied DC, the C=O/(C-C, C-H) ratio decreased as the concentration of the precursor went from 10 to 15%. In this sense, the increase in the as-deposited carbon primarily created during the deposition of C-C/C-H bonds from the precursor structure is probably responsible for the reduction in the number of C=O bonds with precursor concentrations going from 10 to 15%. This hypothesis was also reported by Mertens et al., [32] who observed, while preparing oxygen-rich organic coatings in a DBD at atmospheric pressure, that both the discharge mode and the fragmentation process depend upon the concentration of the precursor in the discharge. In addition, this result is consistent with the O/C ratio from the XPS survey investigation (Figure 3a), in which an increase in applied DC resulted in a decrease in surface-coated oxygen.

Of note, for a given concentration of the precursor, the C=O/C-C, C-H ratio generally drops with the applied DC, most likely due to an increase in the electron collisions in the discharge since the shift in the duty cycle corresponded with a power increase [32]. When this occurs, these highly active species (i.e., electrons) may accelerate the fragmentation of the precursor [19].

### 3.2. Wetting Behavior of Plasma Treated ETFE Samples

To assess the wetting features of plasma-treated surface functionalities, water contact angles (WCAs) were measured on ETFE samples before and following plasma treatments (Figure 6). After 180 s of plasma treatment, ETFE samples became (super)hydrophilic, as supported by the observed decrease in the water contact angle from 92° to <10°. This suggests that the treatment time is sufficiently long for the plasma deposited coating to cover the entire surface with a thin hydrophilic layer. The hydrophilic nature of plasma treated ETFE samples substantiates the formation of hydrophilic functionalities such as C-O, O=C-O and C-N (see Figure 4b). Interestingly, hydrophilicity of these samples can be finely tuned by controlling the amount of the precursor injected in the discharge as well as the applied DC. In general, decreasing levels of DC results in polymers with enhanced wettability, except for the coatings prepared under a concentration of precursor in the discharge of 15%. For instance, at 10% DC, the measured WCAs were slightly lower, most likely due to the high O/C ratio found here (Figure 3a). Superhydrophilic coatings were obtained under 30 and 50% DC and a precursor concentration of 15%.

### 3.3. Surface Energy of Plasma Treated ETFE Samples

Figure 7a shows the total surface energy as a function of the precursor concentration for each applied DC. For all the experimental conditions explored in this study, the total surface energy increased from 24 (untreated sample) to 75–80 dyn/cm, thus confirming the presence of hydrophilic functionalities on the coating. Further understanding of the wetting properties of the plasma treated surface can be drawn from the analysis of the polar surface energy and the dispersive surface energy (Figure 7b,c). Following plasma treatment, the dispersive component of surface energy went from 20 dyn/cm (untreated sample) to 35–45 dyn/cm, reaching a maximum at 45 dyn/cm for a 50% DC applied (Figure 7b). Similarly, the polar component of surface energy increased from 5 (untreated sample) to 35–45 dyn/cm following plasma treatment. Interestingly, the values for 50% DC (35 dyn/cm) were lower than those for lower DCs (e.g., 10%, 30%). From Figure 7c, it can be inferred that the amount of the precursor injected in the discharge does not significantly affect the values of the polar component of surface energy, in contrast to what is found with the dispersive counterpart. That said, these data reveal that increasing levels of DC only entail a slight increase in total surface energy, which is mainly due to an increase in the dispersive surface energy (Figure 7b). As a result, rising levels of applied DC (dissipated power in the discharge) are not always accompanied by an increase in surface polarity, as the XPS survey confirmed less oxygen on the surface using the 50% DC.

### 3.4. FTIR

The study of the chemistry and the structure of the plasma-treated ETFE surfaces is key to gaining insight into the mechanisms behind thin film growth. For this reason, IR features have been compared with those of the untreated ETFE surface and discussed, in terms of band broadening and frequency shifts. Figure 8 shows the FTIR spectra of the ETFE film prior to and after a 180 s plasma treatment. The spectrum of ETFE film reveals several strong bands in the 500–1400 cm^−1^ range that are assigned to stretching and bending modes of CF_2_ groups, as well as a minor band at 2915 cm^−1^ that are due to the asymmetric and symmetric stretching vibrations of CH_2_ and CH_3_ groups. Considering that the depth of penetration of an ATR-FTIR analysis is typically 2 µm, the presence of the ETFE spectral features between 1000 and 1400 cm^−1^ suggests that the thickness of the coatings is most likely less than 2 µm. Asymmetric (νa) and symmetric (νs) stretching of C-H in CH_2_ and CH_3_ groups were observed in the 2870–2970 cm^−1^ range for both plasma-treated and untreated polymers [25]. The wavenumbers at which these vibrations occurred were similar to those reported in hydrocarbon compounds [37]. The presence of plasma-deposited coating is further confirmed by the strong and unresolved band at 3300 cm^−1^ arising from the stretching vibrations of O-H groups [38]. This feature is not observed in the untreated surface. N-H stretching vibrations (ν)in NH_2_ and -NH- groups also contributed to the feature located at 3300 cm^−1^ [25]. The presence of the N-H bonds was further confirmed by the absorption at 1552.4 cm^−1^ (bending mode, δ). These hydrophilic functionalities (i.e., OH, NH_2_, -NH-) are at the origin of the low WCA observed in plasma treated surfaces (Figure 6). The coatings exhibited at least two features related to the hydrophilic C=O group between 1500 and 1800 cm^−1^ that deserve mention. The first one at 1710 cm^−1^ is attributed to C=O bonds in ketone functionalities (RC(=O)R′) [26], while the second one at 1658.5 cm^−1^ is due to the stretching of C=O groups in amide functionalities (N-C=O) [39]. Ketone groups are present in chemical structures of the precursor, while the amide groups suggest strong fragmentation of the precursor during the deposition with nitrogen integration from the carrier gas. For these reasons and as explained below in more detail, the authors used the behavior of the C=O bonds as a function of the experimental conditions to explain the fragmentation process of the precursor.

Figure 9 shows the ATR-FTIR spectra of untreated ETFE and plasma-treated ETFE samples for all the experimental conditions investigated in this work. Regardless of the experimental conditions, similar infrared peak absorptions were observed in untreated ETFE and plasma-treated ETFE samples. Some differences in peak intensities are noticed: increasing the DC resulted in more intense IR bands between 1500 and 1800 cm^−1^ and between 2800 and 3300 cm^−1^ when compared to the fluoropolymer related peaks at lower wavenumbers. More specifically, the maximum absorbance value is 0.1 a.u., 0.25 a.u., and 0.75 a.u. for 10% DC, 30% DC, and 50% DC, respectively. Since increasing the DC produced a more powerful discharge, and consequently more available reactive species in the gas phase, it can be assumed that a thicker coating was deposited at higher DC [40]. In addition, regardless of precursor concentration, increasing DC increases the power per molecule for all precursor concentrations. Figure 9a indicates that for 10% DC, the peak intensities (e.g., 1650 cm^−1^) increased from 3% (i.e., 18 gr/h) to 10% (i.e., 60 gr/h) but it drops with 15% (i.e., 91 gr/h). This unexpected trend might be due to the insufficient available power at 10% DC for the activation of the precursor molecules in the discharge. It can be argued that with a 15% precursor concentration in the gas phase, only a portion of the injected molecules are efficiently fragmented and therefore, only a thinner coating can be deposited on the ETFE sample. Surprisingly, Changes observed in the band intensities following plasma treatment at 30% and 50% DC do not follow this trend (Figure 9b,c), suggesting that increasing the power dissipated in the discharge (increasing DC) triggers different fragmentation mechanisms, which in turn, depends on the concentration of the precursor in the gas phase. At higher DCs (i.e., 30% and 50%), thicker coatings were deposited at 3% precursor concentration. In this case, the combination of high DC (high power) with low precursor concentration results in efficient fragmentation of the precursor molecules. Under these plasma operating conditions, the stretching vibration of C≡N in nitrile groups occurs at 2186 cm^−1^ (Figure 9b,c) [41]. Similar results have been reported by Myung and colleagues [42], who observed the presence of nitrile group at 2185 cm^−1^ following surface polymerization of allylamine in an argon atmosphere using a radio frequency (RF) glow discharge operated under a continuous wave (CW). Of note is that the C≡N bond is very stable as its corresponding dissociation energy is relatively high (~887 kJ/mol) which is higher than the other single or double bonds that were formed on the surface [43]. Under high DC (high-power), Myung highlighted that the high relative IR absorption of C≡N/CH ratios compared to the NH/CH ones. This suggested that amine groups turn into nitrile (C≡N) groups. The nitrile functional group may be formed because of the dehydrogenation of the amine group formed on the surface following interaction with photons, metastable, electrons, and dissociated chemical bonds in the plasma phase [44]. Furthermore, because C≡N bonds appear only at high power, fragmentation processes in the gas phase could also contribute to their formation [45]. Thus, imine groups and, to a smaller extent, nitrile groups were formed. In a previous study, Choukourov [46] et al. investigated the effect of the power dissipated in the discharge on the formation of R-C≡N entities during the polymerization of ethylenediamine (EDA) and diaminocyclohexane (DACH) in a tubular reactor with an external RF (13.56 MHz) excitation under pulsed plasma polymerization. The authors observed that the films produced at higher powers contained a significant amount of double- and triple-bonded atoms containing nitrogen. Moreover, the presence of C≡N in the discharge was confirmed by the detection of optical emissions from excited CN fragments by OES.

Figure 10 shows the curve fitting of the band area between 1500 and 1800 cm^−1^. As aforementioned, this region exhibits features due to the stretching of two types of C=O bonds, namely the C=O in the amide bond (N-C=O) at 1660 cm^−1^ and the C=O bond in ketones (R-CO-R`) at 1710 cm^−1^. An additional peak at 1620 cm^−1^ was also used for the curve fitting of this region. The absorbance at 1620 cm^−1^ cannot be unequivocally assigned to a single species, as several groups can contribute to the absorption in this region. This band could be attributed to the C=O-based functionalities, C=C, and/or C=N stretching vibrations [47,48]. However, based on the literature, this band can be more related to the ΔN-H_x_ band deformation mode in amine [44]. The ketone/amide ratio calculated for the analysis of this region was studied to understand the growth mode of the thin layer.

Figure 11 shows the amide (N-C=O)/ketone (R-CO-R`) ratio as a function of the duty cycle for each precursor concentration. Regardless of the DC applied, this figure shows that injecting more precursor (%) in the gas phase results in coatings with lower amide/ketone ratios. Figure 11 indicates that for 10% DC (0.17 W.cm^−2^), the amide/ketone ratio varied from 1.66 to 1 as the concentration of precursor went from 3% to 15%. For 30% DC (0.51 W.cm^−2^) when the precursor concentration increased from 3% to 15%, the ratio of amide/ketone went down from 2 to 1.5. In addition, a similar trend was observed for 50% DC that the ratio decreased from 2.1 to 1.3 with the increase in the precursor concentration up to 15%.

Figure 11 also demonstrates that, for each precursor concentration inside the discharge, raising the DC from 10% to 30% led to a marginal increase in the level of amide/ketone on the surface, demonstrating higher nitrogen incorporation into the coating. However, the patterns changed when the DC was raised from 30% to 50%. For instance, the amide/ketone ratio marginally increased for a 3% precursor. It nearly stayed the same for the 10% precursor. The ratio was found to decrease slightly for the 15% precursor.

It can be argued that increasing the precursor concentration reduces the available power per molecule to react and deposit. As a result, less power per molecule results in less fragmentation of the precursor structure and nitrogen incorporation, leading to an increase in the deposition of C=O ketone produced from the precursor structure on the surface. Likewise, raising the concentration of organic precursors reduces the energy per molecule available for excitation. Here, the additional organic molecules can quench the excited electronic states of nitrogen, as reported in the literature [47]. In this regard, Guerra et al. [49] mentioned adding precursors with the capability of quenching the metastable can destabilize the discharge since the energy is too low to ionize the reactive components. This means that the role played by metastable on the overall excitation processes is less important when the concentration of the organic molecules increases. Moreover, the dissociation energy of the nitrogen molecule (N_2_ → 2N) is 9.8 eV [50]. Because of this high bond dissociation energy, a lower amount of nitrogen atoms could react with the organic fragments on the surfaces or in the gas phase when a high amount of the precursor is injected into the discharge.

## 4. Conclusions

In this study, a thin coating was deposited on fluoropolymer samples using an atmospheric pressure plasma DBD in N_2_ operated at several duty cycles (DC) and with an organic precursor at different concentrations. The results indicate the formation of new hydrophilic functionalities, such as amide, ketone, amine, hydrocarbon, and hydroxyl groups on the fluoropolymer surface. These components resulted from a competition between several processes including the substitution of fluorine and hydrogen atoms from the fluoropolymer surface with precursor or nitrogen species, the plasma fragmentation of the organic precursor, and the incorporation of nitrogen into the resulting plasma polymer. The results show that the surface was fully covered after a 180-s treatment as evidenced by XPS. Furthermore, functional groups were characterized by different applied powers, by changing the duty cycle. In fact, a reduced oxygen incorporation in the coating surface resulted from using a higher DC, which in turn led to a lower polar surface energy. Moreover, while characterizing the effect of the precursor amount, the amount of nitrogen atoms present in the coating was reduced when a higher amount of precursor was introduced into the discharge. With less nitrogen containing chemical groups, a higher retention of the precursor chemistry was observed. Analyzing the coating functionalities further by curve fitting the C=O stretching mode feature of the FTIR spectra of the various investigated coatings enabled the calculation of amide/ketone ratio. These calculations revealed that for any investigated power injected in the discharge, a smaller precursor amount led to more nitrogen integration into the plasma deposited coating, while a larger amount of precursor resulted in a thin film with a chemical structure closer to one of the precursors. Interestingly, only when using a combination of a lower precursor concentration and higher DC, in other words, at the highest energy per molecule, the formation of C≡N (nitrile) was observed on the surface. (i.e., increasing the number of precursors and decreasing the power/molecule). In conclusion, all DC (i.e., applied power) concentrations higher than 15% of the precursors favor partial polymerization and reticulation processes. Therefore, a low precursor concentration (e.g., 3%) and a high DC (e.g., 50% DC) is more effective for the incorporation of the N_2_ and will generate stronger cross-linked moieties (e.g., amide and nitrile) in coatings.

## Figures and Tables

**Figure 1 materials-16-00942-f001:**
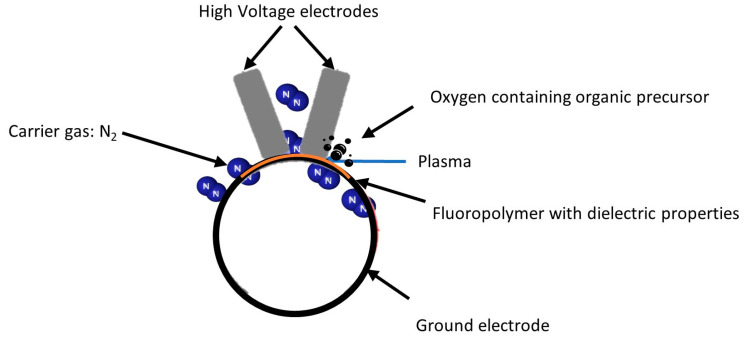
Experimental set-up, semi-industrial (DBD).

**Figure 2 materials-16-00942-f002:**
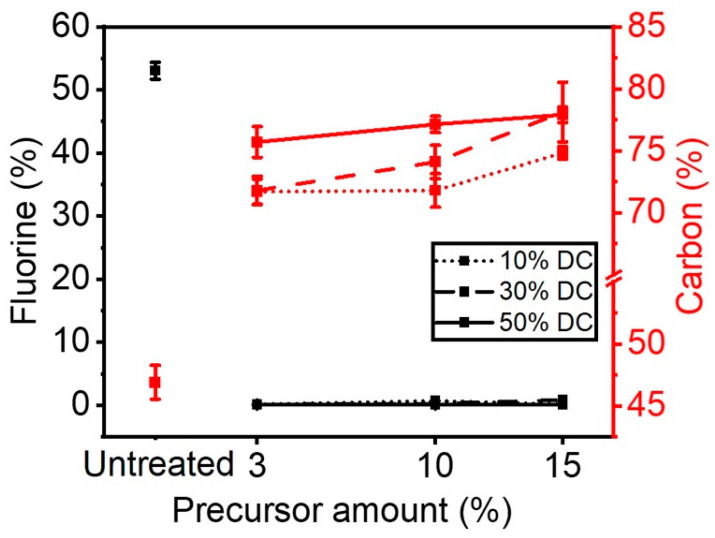
Carbon and fluorine surface concentration as a function of precursor amount and DCs applied.

**Figure 3 materials-16-00942-f003:**
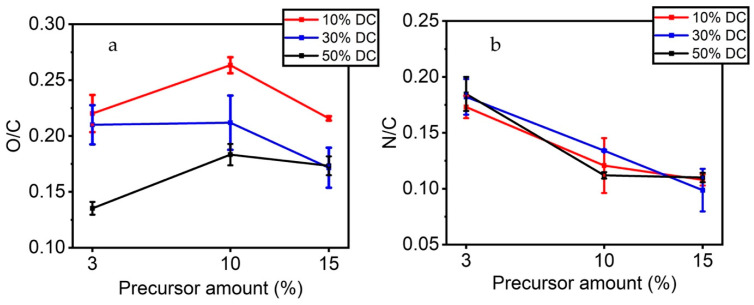
(**a**) O/C and (**b**) N/C ratios of the plasma-coated ETFE samples as function of the precursor amount and applied DC.

**Figure 4 materials-16-00942-f004:**
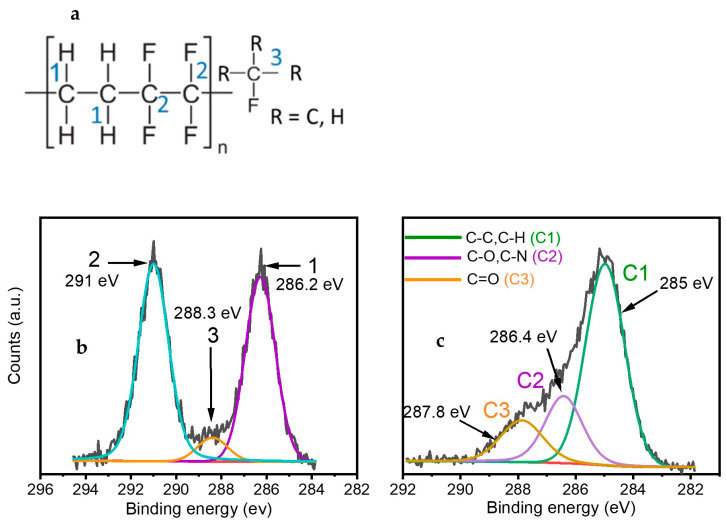
(**a**) Chemical formula of ETFE, and curve fitting of C_1s_ spectrum. (**b**) Untreated ETFE sample. (**c**) Typical plasma treated ETFE sample after a 180 s treatment.

**Figure 5 materials-16-00942-f005:**
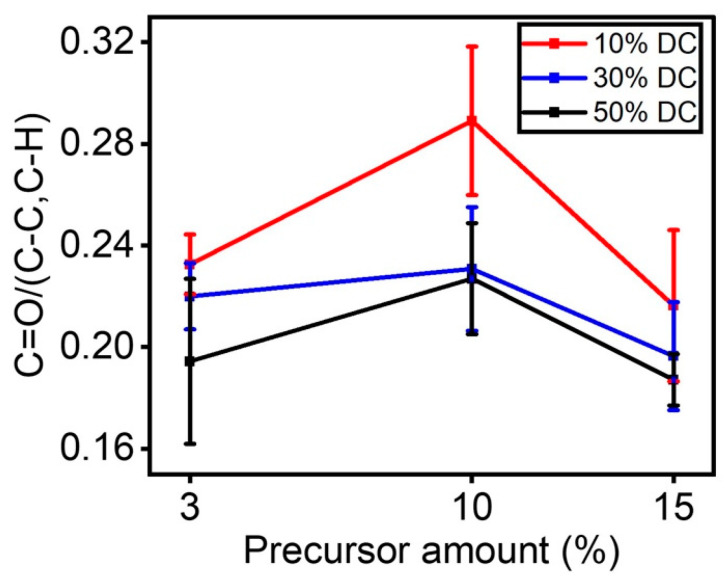
C=O/C-C, C-H ratio as a function of the precursor amount for each DC applied.

**Figure 6 materials-16-00942-f006:**
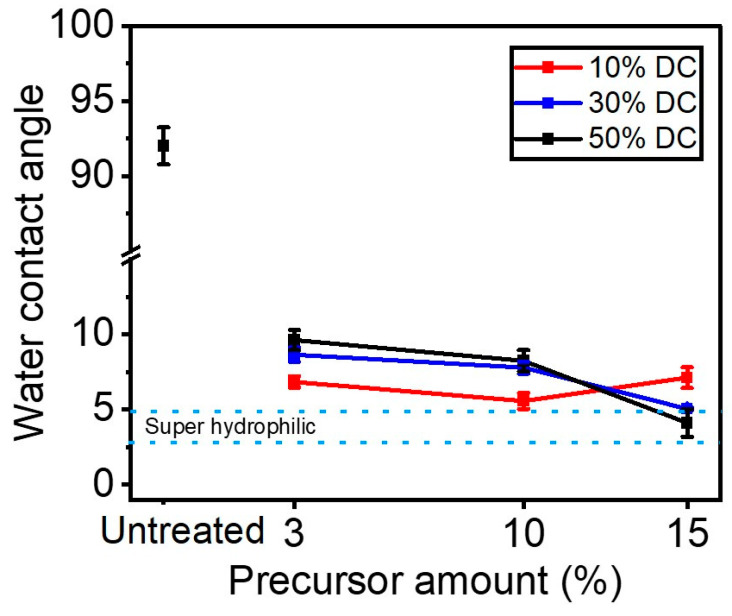
Water contact angle as a function of the precursor (e.g., 3%, 10%, 15%) for different DCs (e.g., 10%, 30%, 50%).

**Figure 7 materials-16-00942-f007:**
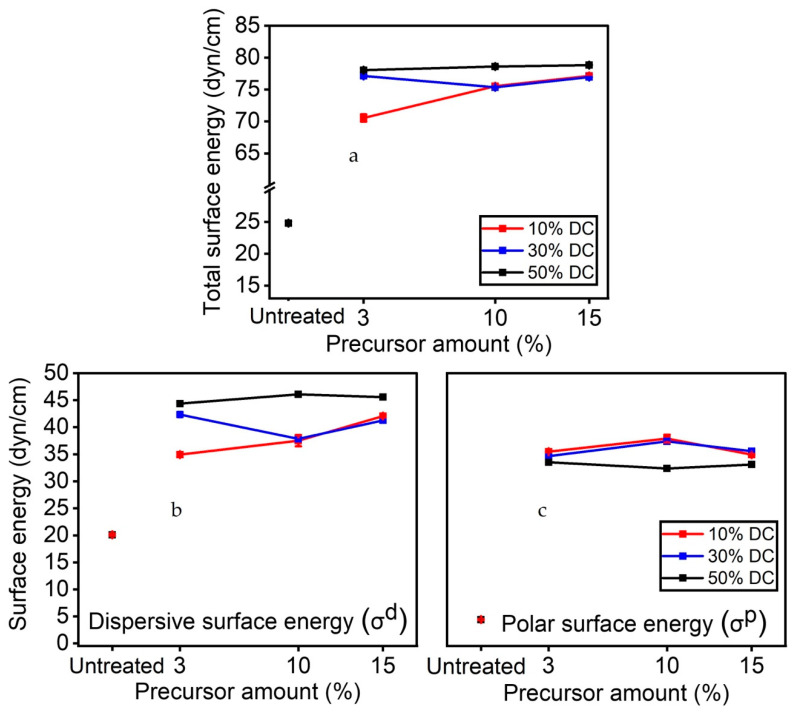
(**a**) Total surface energy; (**b**) dispersive component of surface energy; and (**c**) polar component of surface energy as a function of precursor for each duty cycles.

**Figure 8 materials-16-00942-f008:**
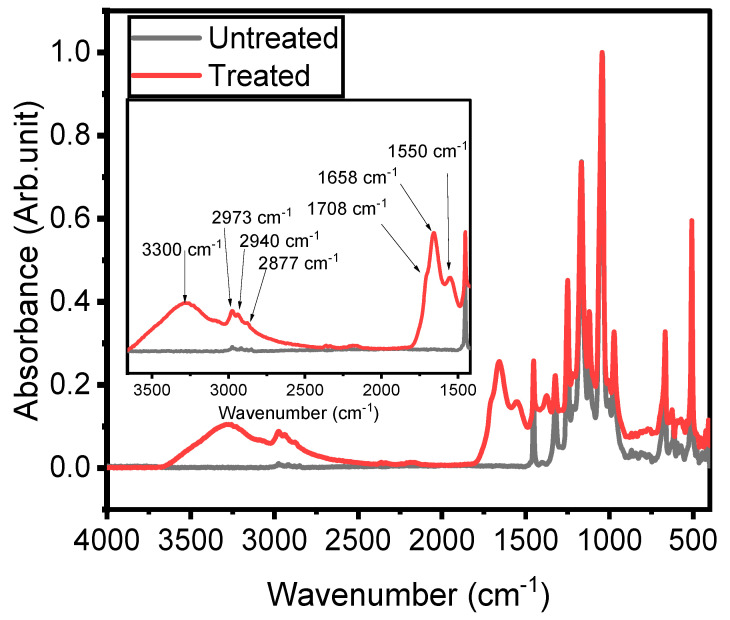
Fourier-transform infrared spectra of untreated ETFE and of a typical plasma-treated ETFE sample.

**Figure 9 materials-16-00942-f009:**
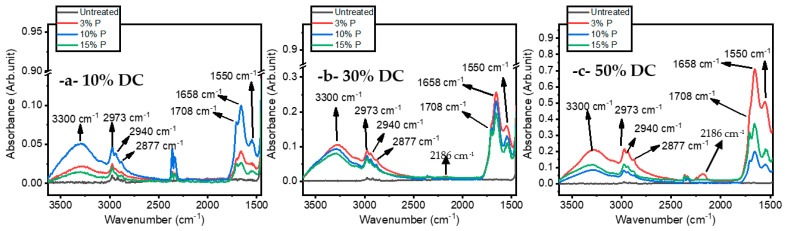
Normalized FTIR absorption spectrum of the ETFE as a function of the precursor concentration (i.e,. 3%, 10%, 15%) for (**a**) 10% DC, (**b**) 30% DC, (**c**) 50% DC.

**Figure 10 materials-16-00942-f010:**
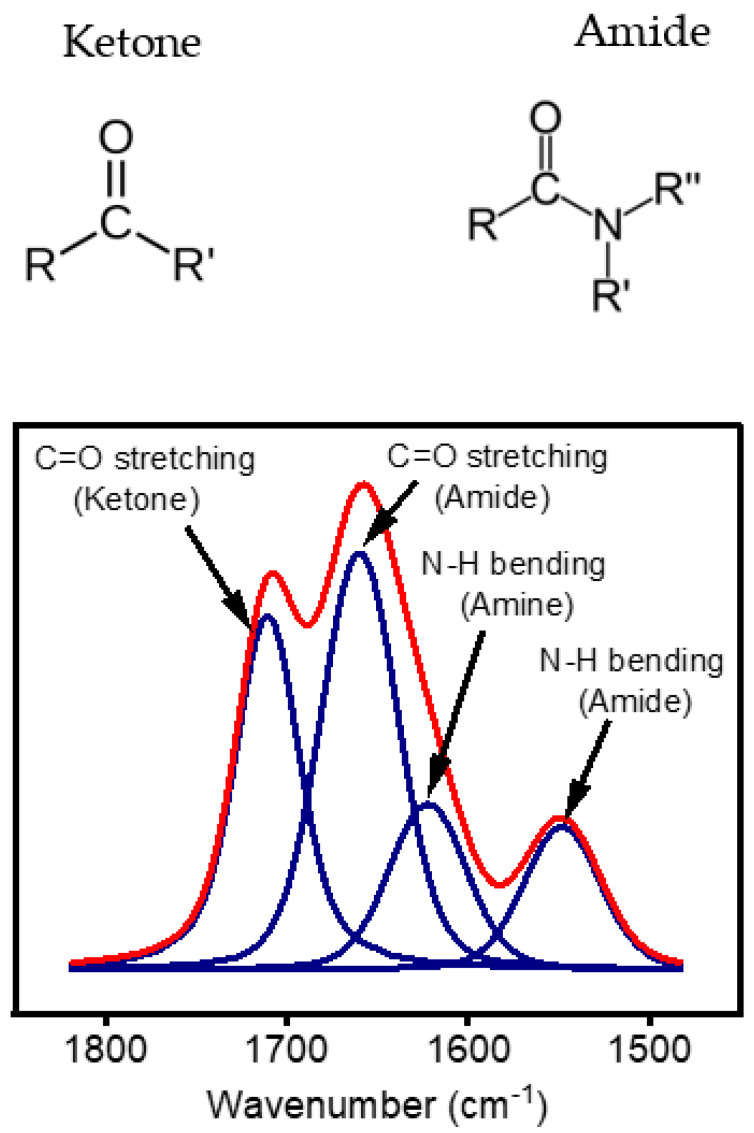
Curve fitting of FTIR spectra in the region between 1500 cm^−1^ and 1800 cm^−1^ to understand the nature of C=O coated on the surface by atmospheric pressure plasma surface treatment.

**Figure 11 materials-16-00942-f011:**
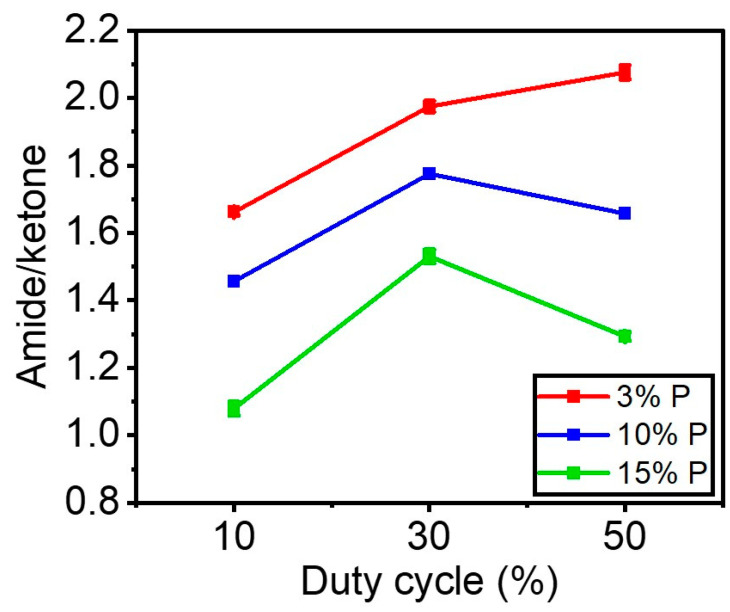
The amide/ketone ratios using the curve fitting under the region (1500–1800 cm^−1^) in FTIR spectra as a function of the duty cycle and amount of precursor.

**Table 1 materials-16-00942-t001:** Plasma experimental conditions.

Voltage (kV)	Frequency (kHz)	Total Gas Flow (SLM)	Gap (mm)	Treatment Duration (s)	Duty Cycle (%)	Power (W.cm^−2^)	Precursor (%)	Precursor Amount (gr.h^−1^)
11	5	5	1	180	10% 30% 50%	0.17 0.51 0.91	3% 10% 15%	18 60 91

**Table 2 materials-16-00942-t002:** Surface composition of plasma-treated ETFE surfaces as a function of applied duty cycle and precursor concentration.

DC (%)	Precursor (%)	285 eV C—C/C—H	286.5 eV C—N/C—O	288.4 eV C=O
10	3	61± 1	23 ± 1	13 ± 1
	10	67 ± 1	13 ± 1	19 ± 1
	15	63 ± 1	20 ± 2	17 ± 1
30	3	68 ± 0.5	14 ± 1	16 ± 0.5
	10	64 ± 2	21 ± 1	15 ± 1
	15	67 ± 1	19 ± 3	13 ± 2
50	3	62 ± 1	26.5 ± 0.5	12 ± 2
	10	69 ± 1	15.5 ± 0.5	16 ± 1
	15	72 ± 2	15± 2	13.5 ± 0.5

**Table 3 materials-16-00942-t003:** Frequencies and assignments of the main absorption peaks found in the IR spectrum of plasma-coated ETFE samples [25,26,36].

Band Position (cm^−1^)	Vibrational Mode	Comment
1550	Stretching, ν N-H	in amine/amide
1660	C=O	in amide
1710	C=O	in ketone
2870–2970	CH_x_	in CH_x_ (x = 2, 3)
3200–3550	O-H	intermolecular bonded
3310–3350	Stretching N-H	in amine/amide

## Data Availability

Data available upon request.

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
