# Peer review of "Fragmentation Mechanism in a Nitrogen Dielectric Barrier Discharge Plasma on Fluoropolymer Polymer Films"

_materials, 2023, doi:10.3390/ma16030942_

Round 1
Reviewer 1 Report
This manuscript described the understanding of the mechanism of the chemical moieties at the fluoropolymer surface. I would recommend the acceptance in its current format.
Author Response
This reviewer recommends the acceptance of this manuscript in its current format. The items for which it is mentioned that they "can be improved" will be, thanks to the comments of the other reviewers.
Reviewer 2 Report
The authors have systematically investigated the deposition of a thin coating on fluoropolymers via the atmospheric pressure plasma dielectric barrier discharge (DBD) by varying duty cycles and different organic precursor concentrations.
The conception, execution and discussion of this work is very good and the experiments are conducted systematically with appropriate discussion. This work is fit for publishing pending these minor changes:
1) The document needs a thorough revision with regards to English language and grammar. Some instances of these issues have been highlighted below:
a) Page 8, Line 306: It has been deemed appropriate to provide further detail on the
b) Page 10: Of note here is that for a given concentration of the precursor
c) Page 91, Line 571: These calculations led to conclude
Author Response
1) A thorough revision of the English language was performed.
a) "It has been deemed appropriate to provide further detail on the..." was changed with "Further details on the carbon bonding environment were assessed using XPS analyses".
b) "Of note here is that for a given concentration of the precursor..." was changed with "Of note, for a given concentration of the precursor...".
c) "These calculations led to conclude..." was changed with "These calculations revealed that...".
Reviewer 3 Report
1.Clearer justification for the choice of experimental conditions
2. It is desirable to give at least a qualitative explanation of the effect of the precursor amount on the final surface properties.
Author Response
- Clearer justification for the choice of experimental conditions: Some sentences were added in page 4, between lines 147 and 155 to clarify the choice of experimental conditions.
- It is desirable to give at least a qualitative explanation of the effect of the precursor amount on the final surface properties: to our point of view, this issue has already been addressed in the current version of the manuscript between lines 230 to 238 and between lines 256 to 270.
Reviewer 4 Report
In this paper, the author investigates the surface modification of fluoropolymers using atmospheric pressure dielectric barrier discharges in a nitrogen and organic precursor environment. It presents interesting data, but there are some points I couldn’t understand. In addition, the way the figures and tables are presented is not correct. Therefore, I labeled this revision as minor. Detailed comments are provided below:
1. Table 1 shows Duty cycle, but 10,30,10%. Is this a misprint? Does it mean that Precursor is 3% when Duty cycle is 10%? Please improve the way the table is presented.
2. Figure 1 shows filled square, circle and triangle. However, it cannot be discriminated. Please make it a little larger. In all graphs below, the notation is small.
3. In figure 4, the spectral data are represented by C1, C2 and C3. However, there does not appear to be a peak or shoulder at the position of C2 in the spectral data. Is it correct to represent it as C2?
4. In Figure 8, the author writes “the spectrum of ETFE film reveals several strong bands in the 1000-1400 cm-1 range”. So what does the peak around 500 cm-1 represent?
Author Response
1. Table 1 shows Duty cycle, but 10,30,10%. Is this a misprint? Does it mean that Precursor is 3% when Duty cycle is 10%? Please improve the way the table is presented: This was indeed a typo. It has been corrected in the revised version of the manuscript.
2. Figure 1 shows filled square, circle and triangle. However, it cannot be discriminated. Please make it a little larger. In all graphs below, the notation is small: This reviewer probably rather refers to Figure 2. It was modified according to this reviewer's requests. In addition, all graphs with small notation were also modified according to this reviewer's requests.
3. In figure 4, the spectral data are represented by C1, C2 and C3. However, there does not appear to be a peak or shoulder at the position of C2 in the spectral data. Is it correct to represent it as C2?: Yes, oxygen and nitrogen were observed in the surface atomic composition; hence, their presence in the C1s spectra was anticipated based on the precursor structure. However, the XPS peak deconvolution depends on several parameters, including the selected function and FWHM. Therefore, the settings must be fair to achieve optimal peak deconversion. In our conditions, the FWHM was consistently between 0.9 and 1.2, requiring the addition of the third peak between 285 eV and 287.5 eV to fit the curves. The presence of this peak made sense because nitrogen had to be a component of a chemical bond. Based on the validation of this bond, the nitrogen and oxygen high-resolution measurements were also performed, although the most identifiable peak for nitrogen and oxygen chemical compositions was 286.5 eV.
4. In Figure 8, the author writes “the spectrum of ETFE film reveals several strong bands in the 1000-1400 cm-1 range”. So what does the peak around 500 cm-1 represent? : This reviewer is right. This sentence was changed with: The spectrum of ETFE film reveals several strong bands in the 500–1400 cm-1 range that are assigned to stretching and bending modes of CF2 groups, as well as a minor band at 2915 cm-1 that are due to asymmetric and symmetric stretching vibrations of CH2 and CH3 groups."
Of note, the band at 510 cm-1 corresponds to CF2 bending, while the band near 690 cm-1 corresponds to the CF band's deformation mode.
Ref: Bracciale, Maria Paola, et al. "Effect of Aging on the Mechanical Properties of Highly Transparent Fluoropolymers for the Conservation of Archaeological Sites." Polymers 14.5 (2022): 912.
Reviewer 5 Report
The reviewed manuscript presents an investigation of the effect of an atmospheric pressure nitrogen DBD with an organic precursor on an ETFE surface at different operating parameters. Oxygen and nitrogen were found to be incorporated into the surface, in which functional groups such as carbonyl and hydroxyl groups formed. Consequently, the surface energy increased. With lower precursor concentration, the nitrogen incorporation was found to increase, and a higher concentration of the precursor leads to a film structure on the surface closer to the one of the precursor. A lower duty cycle increases the oxygen incorporation, and a high duty cycle combined with a low precursor concentration leads to a nitrile formation on the surface.
The manuscript is original, relevant, conclusive and well-written. I recommend publishing it after revising three minor issues that do not require a second review:
Generally
Although an extensive abstract providing context and perspectives is desirable in principle, the abstract in the reviewed manuscript starts with too much general information. Some of it belongs in the introduction rather than in the abstract, e.g. “The highly energetic species…”. In this way, the abstract is unnecessarily wordy. I recommend to slightly change the emphasis of the abstract towards the actual content of your findings, e.g., with a more detailed summary of your conclusions.
2. Experimental section
2.1. Materials
I hope I did not just miss this information, but what “organic oxygen-containing precursor” were you using exactly?
2.2. Plasma reactor
I understood that you are working with a sinusoidal voltage that is switched on and off with a certain duty cycle. First, please refer to the sinusoidal voltage as alternating voltage rather than as alternative voltage. Second and more important, please explain how you realize the duty cycle. Is it a phase-fired controller as in a dimmer switch, or is the sinusoidal signal multiplied with a rectangular signal with a certain frequency and the duty cycle?
Author Response
1- Although an extensive abstract providing context and perspectives is desirable in principle, the abstract in the reviewed manuscript starts with too much general information. Some of it belongs in the introduction rather than in the abstract, e.g. “The highly energetic species…”. In this way, the abstract is unnecessarily wordy. I recommend to slightly change the emphasis of the abstract towards the actual content of your findings, e.g., with a more detailed summary of your conclusions: This sentence was removed from the abstract. In addition, some sentences were added to better summarize our main conclusions.
2- I hope I did not just miss this information, but what “organic oxygen-containing precursor” were you using exactly? : As mentioned to the Editor at the time of submitting the manuscript, this information cannot be disclosed due to a confidentiality agreement signed with the industrial sponsor.
3- I understood that you are working with a sinusoidal voltage that is switched on and off with a certain duty cycle. First, please refer to the sinusoidal voltage as alternating voltage rather than as alternative voltage. Second and more important, please explain how you realize the duty cycle. Is it a phase-fired controller as in a dimmer switch, or is the sinusoidal signal multiplied with a rectangular signal with a certain frequency and the duty cycle?: This was clarified between lines 163 and 165 of the manuscript.